# Long Non-Coding RNA CRYBG3 Promotes Lung Cancer Metastasis via Activating the eEF1A1/MDM2/MTBP Axis

**DOI:** 10.3390/ijms22063211

**Published:** 2021-03-22

**Authors:** Anqing Wu, Jiaxin Tang, Ziyang Guo, Yingchu Dai, Jing Nie, Wentao Hu, Ningang Liu, Caiyong Ye, Shihong Li, Hailong Pei, Guangming Zhou

**Affiliations:** 1State Key Laboratory of Radiation Medicine and Protection, School of Radiation Medicine and Protection, Medical College of Soochow University, Suzhou 215123, China; aqwu115@suda.edu.cn (A.W.); 18862188074@163.com (J.T.); m18769669630_1@163.com (Z.G.); yingchudai@163.com (Y.D.); jingnie@suda.edu.cn (J.N.); wthu@suda.edu.cn (W.H.); liuna@suda.edu.cn (N.L.); caiyongye@suda.edu.cn (C.Y.); shhli@suda.edu.cn (S.L.); 2Collaborative Innovation Center of Radiological Medicine of Jiangsu Higher Education Institutions, Suzhou 215123, China

**Keywords:** long non-coding RNA, eEF1A1, MDM2, metastasis, NSCLC

## Abstract

The occurrence of distant tumor metastases is a major barrier in non-small cell lung cancer (NSCLC) therapy, and seriously affects clinical treatment and patient prognosis. Recently, long non-coding RNAs (lncRNAs) have been demonstrated to be crucial regulators of metastasis in lung cancer. The aim of this study was to reveal the underlying mechanisms of a novel lncRNA LNC CRYBG3 in regulating NSCLC metastasis. Experimental results showed that LNC CRYBG3 was upregulated in NSCLC cells compared with normal tissue cells, and its level was involved in these cells’ metastatic ability. Exogenously overexpressed LNC CRYBG3 increased the metastatic ability and the protein expression level of the metastasis-associated proteins Snail and Vimentin in low metastatic lung cancer HCC827 cell line. In addition, LNC CRYBG3 contributed to HCC827 cell metastasis in vivo. Mechanistically, LNC CRYBG3 could directly combine with eEF1A1 and promote it to move into the nucleus to enhance the transcription of MDM2. Overexpressed MDM2 combined with MDM2 binding protein (MTBP) to reduce the binding of MTBP with ACTN4 and consequently increased cell migration mediated by ACTN4. In conclusion, the LNC CRYBG3/eEF1A1/MDM2/MTBP axis is a novel signaling pathway regulating tumor metastasis and may be a potential therapeutic target for NSCLC treatment.

## 1. Introduction

Lung cancer is a common respiratory system malignancy that is characterized by rapid proliferation and metastasis. Non-small cell lung cancer (NSCLC), in particular, has a high rate of mortality and metastasis [1]. Despite some progress in cancer treatment in the past decades, lung cancer remains a considerable challenge due to poor treatment efficacy and survival rates. The occurrence of distant tumor metastases is a major barrier for lung cancer therapy, and seriously affects clinical treatment and prognosis [2]. A further understanding of the molecular mechanism underlying lung cancer invasion and metastasis may provide a novel target for early diagnostic assessment and treatment of lung cancer patients.

Long non-coding RNAs (lncRNAs) have a length of >200 nucleotides without protein coding capacity and can bind to various molecules to regulate tumor-associated gene expression in tumorigenesis [3]. Many lncRNAs have been reported to play key roles in promoting lung cancer cell migration both in vivo and in vitro, including UPAT [4], ENST457720 [5], HOXD-AS1 [6], and MALAT1 [7]. These lncRNAs are up-regulated in higher grade malignancy NSCLC and their main functions are inducing the epithelial-mesenchymal transition (EMT) and tumor cell migration. Recently, several studies have revealed that lncRNAs can regulate the proliferation, migration, and invasion of multiple types of malignant tumor cells by influencing murine double minute 2 (MDM2) expression [8]. MDM2 is an important E3 ubiquitin ligase driving p53 degradation, which also forms an auto-regulatory feedback loop with p53 [9]. However, there is evidence supporting the notion that MDM2 is an oncogene with both p53-dependent and -independent oncogenic activities. It also often has increased expression levels in a variety of human cancers [10]. Emerging studies have revealed that MDM2 promotes tumor cell metastasis in a p53-independent manner. For example, MDM2 was found to be an E3 ligase targeting epithelial marker E-cadherin for ubiquitin-dependent degradation that also promoted cell invasiveness [11]. Another study has shown that MDM2-induced cell migration is associated with an increase in vascular endothelial growth factor expression [12]. 

Recent findings have demonstrated that MDM2-binding protein (MTBP) is a novel metastasis suppressor inhibiting cell migration. MTBP overexpression in human osteosarcoma cells lacking wild-type p53 inhibits metastasis, and MTBP acts as a metastasis suppressor independent of the p53 status. Furthermore, the function of MTBP in suppressing cell migration and filopodia formation is related to inhibiting the function of an actin crosslinking protein α-actinin-4 (ACTN4) [13]. ACTN4 was first identified as an actin-binding protein related to cell motility and metastasis in breast cancer. Immunofluorescence studies using SaOs2-LM7 cells revealed a partial co-localization of MTBP and ACTN4, mainly in the cytoplasm. In addition, the carboxyl (C)-terminal region of MTBP (amino acid 760 to 904) is required for the MTBP-ACTN4 interaction. Interestingly, MDM2 also binds to a 380-amino acid region at the C-terminal MTBP, which contains the ACTN4 binding domain [13,14]. Therefore, we hypothesized that MDM2 might influence the interaction between MTBP and ACTN4 in modulating cell migration.

Our previous study identified a novel lncRNA LNC CRYBG3 that was up-regulated in lung cancer and inducible by ionizing radiation [15]. The present study found that LNC CRYBG3 can combine with eukaryotic translation elongation factor 1 alpha (eEF1A1), which is not only a translation factor, but also a pleiotropic protein that is highly expressed in human tumors, including breast, ovarian, and lung cancers. eEF1A1 modulates cytoskeleton assembly, cell proliferation, and cell death. It also serves as a molecular chaperone for some transcription factors [16]. eEF1A1 was identified as a binding partner for LNC CRYBG3 in the present study. LNC CRYBG3 and eEF1A1 can cooperate to combine in the MDM2 gene promoter region and enhance MDM2 expression. MDM2 overexpression then promotes the migration and invasion of lung cancer cells by interacting with MTBP. These findings provide new insights into the functional link among LNC CRYBG3, eEF1A1, MDM2, and MTBP in lung cancer progression.

## 2. Results

### 2.1. LNC CRYBG3 Is Upregulated in NSCLC Cells

NSCLC is a malignant tumor with high rates of recurrence and metastasis. In our previous study, LNC CRYBG3 was found to be highly expressed in the lung cancer tissues compared to normal lung tissues [15]. Similarly, LNC CRYBG3 levels were also determined in several lung cancer cell lines, including HCC827, A549, H460, and H1299, and in the normal lung cell lines Beas-2B and HSAEC1-KT. The results were the same as those for the clinical sample analysis, where LNC CRYBG3 was highly expressed in lung cancer cell lines compared to normal lung cell lines (Figure 1A). Intriguingly, the HCC827 cell line had the lowest LNC CRYBG3 expression in the four assessed lung cell lines. Its migration and invasion activities were markedly the weakest relative to the other three cell lines, which were analyzed using a Transwell assay (Figure 1B–E). Western blot analysis revealed that the protein levels of migration-dependent proteins Vimentin and Snail were significantly low in HCC827 cells; in addition, the expression of epithelial marker E-cadherin was significantly high but that of the mesenchymal marker N-cadherin was significantly low in HCC827 cells (Figure 1F). Compared to the A549, H460, and H1299 cells, these observations suggested that the low level of LNC CRYBG3 may be the main reason for the weak metastatic ability of HCC827 cells. Thus, the HCC827 cell line was selected as a low migration cell model for studying the role of LNC CRYBG3 in modulating metastasis compared with the A549 cell line in the following experiments.

### 2.2. LNC CRYBG3 Promotes NSCLC Cell Metastasis

To further investigate the function of LNC CRYBG3 in NSCLC cells, LNC CRYBG3 was knocked down using shRNA lentivirus particles in A549 cells (A549 shLNC CRYBG3 cells) and overexpressed in HCC827 cells using LNC CRYBG3 adenovirus particles (HCC827 LNC CRYBG3 cells) (Figure 2A and Appendix A). Transwell assays indicated that compared with the cells treated with control shRNA, LNC CRYBG3 suppression attenuated the invasive and migratory abilities of A549 cells; conversely, the migratory and invasive abilities of HCC827 cells with overexpressed LNC CRYBG3 were enhanced compared with the negative control (Figure 2B–E). In order to confirm the correlations between LNC CRYBG3 and EMT-related genes, Western blotting was conducted and the results showed that overexpression of LNC CRYBG3 increased the level of mesenchymal marker N-cadherin and Vimentin as well as the transcription factor Snail in HCC827 cells. Furthermore, the expression of these EMT-related proteins was suppressed when LNC CRYBG3 expression was interfered in A549 cells (Figure 2F). These results indicate that sufficient LNC CRYBG3 expression promotes the metastasis of NSCLC cells.

### 2.3. LNC CRYBG3 Enhances NSCLC Cell Metastatic Ability in NOD/SCID Mice

To determine whether LNC CRYBG3 can promote tumor metastasis in vivo, A549 shLNC CRYBG3, HCC827 LNC CRYBG3, and negative control cells with stable luciferase expression were established. These cells were intravenously injected into NOD/SCID mice. Then, the luciferase signal was examined using an imaging system to monitor the locations and growth of tumor metastatic foci. Six weeks after the injection, A549 shLNC CRYBG3 cells produced significantly weaker pulmonary metastasis relative to negative control cells. On the contrary, more HCC827 LNC CRYBG3 cells migrated to the lung compared with the control (Figure 3A). Anatomical observation of these mouse lungs revealed fewer tumorous nodules in mice injected with A549 shLNC CRYBG3 cells than with the control. The number of nodules in mice injected with HCC827 LNC CRYBG3 cells was increased compared with the control (Figure 3B,C). In addition, it was confirmed that the lung nodules were generated from intravenously injected tumor cells using immunohistochemical examination of cell proliferation marker Ki67 and hematoxylin and eosin staining of pulmonary pathological sections (Figure 3D,E). Immunohistochemical assays showed that the effects of LNC CRYBG3 on the expression of migration-dependent proteins Vimentin and Snail and EMT markers N-cadherin and E-cadherin were the same as those demonstrated using in vitro assays (Figure 3F). These results indicate that overexpressed LNC CRYBG3 can promote NSCLC metastasis in vivo.

### 2.4. LNC CRYBG3 Combines with eEF1A1 and Increases Its Nucleus Translocation

It has been reported that lncRNA is involved in the regulation of tumor progression by modulating DNA methylation, functional protein modification, and microRNA function. To further investigate the potential direct binding between LNC CRYBG3 and its target proteins, an RNA pull-down assay was performed using biotin-labeled LNC CRYBG3 and antisense-LNC CRYBG3 control. The biotin labeled probe was shown to be correct using agarose gel electrophoresis and the target band at 300 bp was confirmed as LNC CRYBG3. LNC CRYBG3 complexes captured on the beads were detected using SDS-PAGE silver staining (Appendix A). Then, the target band was excised and detected using mass spectrometry. Mass spectrometry results showed an eEF1A1 peptide with a sequence identified as LPLQDVYK, suggesting that LNC CRYBG3-combined extractions possibly contain eEF1A1 (Appendix A). To further verify the direct combination of LNC CRYBG3 with eEF1A1, RNA pull-down and immunoprecipitation experiments were performed and the results showed that eEF1A1 was pulled down by LNC CRYBG3, while LNC CRYBG3 was precipitated by eEF1A1 (Figure 4A,B). When evaluating the effect of LNC CRYBG3 and eEF1A1 interaction, immunofluorescence assay was conducted and the results showed that eEF1A1 was predominantly located in the nuclei when LNC CRYBG3 was overexpressed (Figure 4C,D). These findings imply that LNC CRYBG3 targets eEF1A1 to increase eEF1A1 translocation to the nuclei.

### 2.5. LNC CRYBG3 Mediates the Binding of eEF1A1 to the Promoter of MDM2 Gene

eEF1A1 has been demonstrated to be a transcription factor chaperone in the nucleus [17]. Therefore, we tested whether eEF1A1 functions as a transcription factor chaperone affecting LNC CRYBG3-induced MDM2 expression or not. A549 shLNC CRYBG3, HCC827 LNC CRYBG3, and negative control cells were transfected with a plasmid containing a luciferase reporter gene in the constitutive wild-type or mutant MDM2 promoter region. The luciferase assays showed that increasing LNC CRYBG3 enhanced, while silencing LNC CRYBG3 attenuated, the luciferase reporter activity of the wild-type MDM2 promoter, but not of the mutant form (Figure 5A–C). However, when eEF1A1 expression was inhibited by small interference RNA (siRNA) in HCC827 LNC CRYBG3 cells, the LNC CRYBG3-increased luciferase reporter activity of wild-type MDM2 promoter was attenuated (Figure 5D). Analogously, immunoblot analysis reveal that silencing eEF1A1 reduced the LNC CRYBG3-induced MDM2 expression. In addition, immunoblot and Transwell assays confirmed that suppression of eEF1A1 or MDM2 by siRNA abolished the migration and invasion stimulated by LNC CRYBG3 overexpression in HCC827 LNC CRYBG3 cells (Figure 5E–I). Therefore, these results indicate that LNC CRYBG3 promotes the metastasis of NSCLC cells by interacting with eEF1A1 to upregulate MDM2 expression.

### 2.6. MDM2 Promotes Metastasis by Blocking the MTBP and ACTN4 Interaction

The mechanism of extensive MDM2 expression in controlling cell migration remains elusive. It has been previously reported that MTBP suppressed cell migration and filopodia formation by inhibiting ACTN4 [13]. Importantly, the domain for MTBP combining with MDM2 contained a region with a common binding site of ACTN4 [14]. Immunofluorescence and co-immunoprecipitation assays were performed to determine whether the increased MDM2 expression induced by elevated LNC CYBG3 can inhibit the combination of MTBP and ACTN4 in NSCLC cells. Immunofluorescence experiments showed that LNC CRYBG3 overexpression enhanced MTBP and MDM2 co-localization in HCC827 cells, but reduced the interactions between MTBP and ACTN4 (Figure 6A–D). Furthermore, co-immunoprecipitation showed that the combination of MTBP and MDM2 was increased in HCC827 cells with overexpressed LNC CRYBG3 and decreased in LNC CRYBG3-knockdown A549 cells. Conversely, MTBP and ACTN4 binding was reduced in elevated LNC CRYBG3 HCC827 cells and increased in A549 cells with depleted LNC CRYBG3 (Figure 6E and Appendix A). Western blotting showed that there was no LNC CRYBG3 influence on the expression of MTBP and ACTN4 (Figure 6F). These observations demonstrated that the inhibitory effect of MDM2 on the metastasis suppressor MTBP is the main reason for LNC CRYBG3-induced NSCLC cell metastasis.

## 3. Discussion

Many previous studies have indicated that lncRNAs perform important functions in carcinogenesis and cancer progression [18,19,20,21,22]. Herein, we revealed that LNC CRYBG3 expression was remarkably increased in patient lung cancer tissues, particularly in metastatic tumors. It was also related to NSCLC prognosis. More elevated LNC CRYBG3 expression was found in lung cancer cell lines than in normal cell lines. In addition, our previous study suggested that LNC CRYBG3 is a regulator of glycolysis and that its overexpression promotes lung cancer cell proliferation [15]. The present study demonstrated that LNC CRYBG3 overexpression promotes migration and invasion abilities of less migrated HCC827 cells and LNC CRYBG3 silencing inhibits the metastasis of highly migrated A549 cells. Furthermore, alteration of LNC CRYBG3 expression can influence epithelial-like morphological features in these two cell types. For example, N-cadherin was downregulated and E-cadherin was upregulated in LNC CRYBG3-knockdown A549 cells. LNC CRYBG3 increase also enhanced the expression of Snail and Vimentin, which are necessary for NSCLC cell migration. A distant tumor metastasis is the main concern for lung cancer treatment, and may cause tumor relapse and seriously influence patient survival in clinical practice. Identifying the molecular mechanism of tumor cell migration is very helpful for designing an effective treatment plan for a lung cancer patient. However, there have been not enough reports on the role of lncRNAs in tumor metastasis. Many earlier studies have found EMT to be an important process for tumor metastasis, which is regulated by many lncRNAs. LncRNA MALAT1 has been found to be upregulated in multiple cancers, including lung cancer and hepatoma. It also promotes distant tumor metastasis by enhancing EMT [23,24,25,26]. Here, LNC CRYBG3 was found to significantly promote EMT. Its expression level was involved in Snail and Vimentin expression in NSCLC cells and tissues.

LNC CRYBG3 can bind to eEF1A1, which is not only a translation factor, but also a pleiotropic protein that is highly expressed in human tumors, including breast, ovarian, and lung cancers [16,27,28]. eEF1A1 has multiple regulatory functions during cancer progression, including modulating the cytoskeleton, exhibiting chaperone-like activity, and regulating cell proliferation and cell death [16]. Few reports have discussed the role of eEF1A1 in tumor metastasis. The present study found that LNC CYRBG3 can directly bind eEF1A1 and promote eEF1A1 movement into the nucleus, although it did not influence its expression. However, LNC CYRBG3 overexpression can markedly increase MDM2 levels. Many previous studies have demonstrated that aberrant MDM2 expression plays an important role in proliferation and migration of lung cancer cells [29,30,31]. Therefore, we speculated that eEF1A1 may be involved in LNC CYRBG3 promoting MDM2 expression. As expected, our results further revealed that eEF1A1 depletion significantly inhibited LNC CYRBG3-induced MDM2 expression and cell migration and invasion. Moreover, luciferase reporter assays showed that eEF1A1 directly targets the MDM2 promoter region in lung cancer cells and that LNC CRYBG3 has a high affinity for the binding of eEF1A1 to the MDM2 promoter. A recent study revealed that eEF1A1 can recruit HSF1 to the HSP72 gene promoter to initiate transcription. Subsequently, it stabilizes and transports HSP72 mRNAs to the translating ribosomes by binding to their 30 untranslated regions [17]. Therefore, we speculated that eEF1A1 may serve as a transcription factor chaperone participating in MDM2 transcription regulation and that the binding of LNC CYBG3 and eEF1A1 may enhance eEF1A1 transport from the cytoplasm to the nucleus to regulate MDM2 expression.

The present study found that LNC CRYBG3 overexpression increased MDM2 levels and led to lung cancer cell metastasis, while LNC CRYBG3 depletion suppressed MDM2 expression and metastasis. The classical function of MDM2 is p53 suppression. It plays very important roles in the p53 related pathways. However, many recent publications strongly suggest that MDM2 promotes tumorigenesis via p53-independent mechanisms [10]. Similarly, the present results showed that the effects of MDM2 alteration on metastasis in wild-type, null, or mutational p53 cells were the same, suggesting that a potential mechanism for LNC CRYBG3 in regulating metastasis is not related to p53 status (Appendix A). While attempting to elucidate the mechanisms behind p53-independent roles of MDM2 in tumorigenesis, MTBP was identified as a novel target that interacts with MDM2 [32]. Intriguingly, it has been reported that MTBP suppresses cancer metastasis partly via inhibition of ACTN4 function [13]. Our observations showed that LCN CRYBG3-induced MDM2 overexpression was able to reduce the binding of MTBP to ACTN4. Therefore, we hypothesized that MDM2 promotes cancer cell migration by interacting with MTBP to block the MTBP and ACTN4 binding. These findings can also explain why MDM2 overexpression is frequently associated with cancer metastasis and worse prognosis in cancer patients independent of p53.

In conclusion, high LNC CRYBG3 expression promotes migration of NSCLC cells both in vitro and in vivo by activating the eEF1A1-MDM2-MTBP axis. Its underlying mechanism involves LNC CRYBG3 inducing MDM2 expression by accelerating eEF1A1 movement into the nucleus to bind to the promoter region of the MDM2 gene. Then, elevated MDM2 actively inhibits MTBP-mediated metastasis suppression by blocking the interaction of MTBP and ACTN4 (Figure 7). Based on these results, LNC CRYBG3 may be a potential target for lung cancer prevention and treatment in the clinic.

## 4. Materials and Methods

### 4.1. Cell Culture

A549, H460, H1299, and HCC827 cells were cultured in RPMI-1640 medium (Sigma, St. Louis, MO, USA). HSAEC1-KT and BEAS-2B cells were cultured in DMEM (Sigma). Both RPMI-1640 and DMEM were supplemented with 10% fetal bovine serum (Gibco, Grand Island, NY, USA), 1% penicillin sodium, and 100 µg/mL streptomycin at 37 °C in 5% CO_2_ in a humidified incubator (Thermo Fisher Scientific, Asheville, NC, USA).

### 4.2. RNA Isolation and Real-Time Quantitative Polymerase Chain Reaction (RT-qPCR)

Total RNA was extracted using Trizol reagent (Invitrogen, Carlsbad, CA, USA). RNA was reverse transcribed into cDNA with PrimeScript RT Reagent Kit (Takara, Tokyo, Japan). RT-qPCR was performed with PowerUp™ SYBR^®^ Green Master Mix (Life Technologies, Grand Island, NY, USA) in a Life Technologies system (Thermo Fisher Scientific). The primers used for LNC CRYBG3 were as follows: forward: GAAGAGTGGAAGTGCGAAGGA; reverse: GGCAATGACCCCATTAGCTC. The primers for GAPDH were as follows: forward: GCACCGTCAAGGCTGAGAAC; reverse: TGGTGAAGACGCCAGTGGA. GAPDH was used as a normalization control. The average of the three independent analyses for each gene was calculated. The 2^–ΔΔ*C*t^ method was used to determine the relative gene expression levels.

### 4.3. Gene Silencing Overexpression and Reporter Plasmids

A549 cells were transfected with LNC CRYBG3 shRNA lentivirus particles (Sangon, Shanghai, China) for 24 h and then screened with medium containing 2 g/mL of puromycin (Invitrogen) and double-checked with RT-qPCR. HCC827 cells were transfected with LNC CRYBG3 adenovirus particles with EGFP reporter (Sangon) for LNC CRYBG3 overexpression. To construct a plasmid examining the combination of eEF1A1 and the MDM2 promoter region, the MDM2 promoter region and luciferase reporter gene sequence were inserted into the pcDNA3.1 vector (Sangon). The plasmids were transfected into cells using Lipofectamine 3000 Reagent (Invitrogen). Luciferase activity was measured 48 h later using the Luciferase Reporter Assay System (Promega, Madison, WI, USA) with Multiscan Spectrum (BioTek, Burlington, VT, USA).

### 4.4. Transwell Assay

Cell migration and invasion were examined using 24-well Boyden chambers (6.5-mm diameter, 8-μm pores; Corning, NY, USA). The upper chambers coated with or without Matrigel (BD Biosciences, San Jose, CA, USA) were used for detecting invasion or migration, respectively. After starving in serum-free medium for 6 h, a total of 40,000 cells/well were seeded in the upper chambers and cultured in the medium without serum. A total of 600 μL of medium with 10% FBS was added to the bottom chamber. Cells were continuously cultured for 12 h (migration) or 24 h (invasion). Then, non-migrating cells and Matrigel in the upper chamber were removed using a cotton swab. The filters with invaded cells were independently fixed with 4% paraformaldehyde, stained with 0.1% crystal violet, and captured using a light microscope (Olympus, Tokyo, Japan). The stained cells were photographed and quantified by counting in 6 random fields (200×). Graphs showed relative cell migration or invasion (%) compared to the number of migrating cells in control (shNC or NC).

### 4.5. Animal Experiment

A total of 2 × 10^6^ A549-shLNC CRYBG3, HCC827-LNC CRYBG3, and negative control cells stably transfected with luciferase reporter gene were intravenously injected into the tails of six-week-old NOD/SCID mice (5 mice each group). Six weeks after the tumor cell injection, all mice were intraperitoneally injected with luciferase substrate and then imaged using an in vivo animal imaging system (PerkinElmer, Waltham, MA, USA). All mice were supplied by the SPF Animal Laboratory, Soochow University (Suzhou, China). All animal studies were reviewed and approved by the Soochow University Institutional Animal Care and Use Committee.

### 4.6. Western Blot Assay

Cell protein samples were extracted using RIPA buffer. Total protein concentration was quantified using a DC Protein Assay Kit I (Bio-Rad, Los Angeles, CA, USA). Then, the samples were denatured for 5 min at 100 °C. Total protein samples were separated using 10% sodium dodecyl sulfate polyacrylamide gel electrophoresis and then transferred onto polyvinylidene difluoride membrane (Millipore, Boston, MA, USA). The membranes were immersed in 5% nonfat milk to block heterogenetic antigens for 1 h at room temperature and then hybridized with primary antibodies in 5% bovine albumin (BSA) in 1× phosphate-buffered saline with Tween detergent (PBST) overnight at 4 °C. After washing three times with PBST, membranes were incubated with horseradish peroxidase-conjugated secondary antibody at room temperature for 2 h. Signal visualization was performed using an ECL kit (Millipore), recorded using a polychromatic fluorescence chemiluminescence imaging analysis system (ProteinSimple, San Francisco, CA, USA) and quantified by densitometry (Image J). Antibodies used in western blotting were: N-cadherin, Snail, Vimentin, MDM2, ACTN4 and GAPDH (Abcam, Cambridge, UK); E-cadherin, eEF1A1 and MTBP (Santa Cruz Biotechnology, Santa Cruz, CA, USA).

### 4.7. siRNA Transfection

Small interference RNAs (siRNAs) were purchased from Sangon Biotech, Shanghai, China. siRNA sequences were as follows: for non-specific control: sense, 5′-UUCUCCGAACGUGUCACGUTT-3′; antisense, 5′-ACGUGACACGUUCGGAGATT-3′; for anti-eEF1A1: sense, 5′-CCCAGGACACAGAGACUUUAUTTT-3′; antisense, 5′-AUAAAGUCUCUGUGUCCUGGGTT-3′; and for anti-MDM2: sense 5′-GGAAGAAACCCAAGACAAATT-3′; antisense, UUUGUCUUGGGUUUCUUCCTT-3′. A total of 20 pmol of each siRNA were used for transfection in six-well dishes performed with the RNAi-Mate Kit (Gene Pharma, Shanghai, China) according to the manufacturer’s instructions.

### 4.8. Co-Immunoprecipitation, RNA Pull Down and Immunoprecipitation Analysis, Mass Spectrometry

Cells were lysed with immunoprecipitation dedicated cell lysis buffer (Beyotime, Shanghai, China). Approximately 50 μL of total cell lysates were incubated with protein-specific antibodies or their matched isotype antibodies overnight at 4 °C. Then, the antibody-protein complex was incubated with protein A/G plus-agarose beads (Bimake, Shanghai, China) at room temperature for 2 h. Protein-coated beads were separated and boiled in a loading buffer. Precipitated protein complexes were separated for western blot assay. For RNA pull down and immunoprecipitation analysis, and mass spectrometry, refer to our previous publication [15].

### 4.9. Immunofluorescence

Cells were cultured on glass cover slips in a 24-well plate. Cells were then fixed with 4% paraformaldehyde for 15 min, washed with PBS three times, permeabilized with 0.2% Triton-X 100 at room temperature for 15 min, blocked in 5% BSA for 1 h, and incubated with target protein-specific antibodies at room temperature for 2 h. After washing with PBST three times, the fluorescent second antibody was added for incubation at room temperature for 1 h in a light-free environment. The samples were then washed with PBST three times and mounted using the anti-quenching reagent with DAPI (Beyotime). Finally, fluorescent images were captured using confocal microscopy (Olympus).

### 4.10. Statistics

Statistical analysis was performed using GraphPad Prism 8 (GraphPad Software Inc., San Diego, CA, USA). All experiments were independently repeated at least three times and all data were represented as means ± standard deviation. Two tailed Student’s *t*-tests were used for statistical analysis. Differences were considered significant if * *p* < 0.05, ** *p* < 0.01, and *** *p* < 0.001.

## Figures and Tables

**Figure 1 ijms-22-03211-f001:**
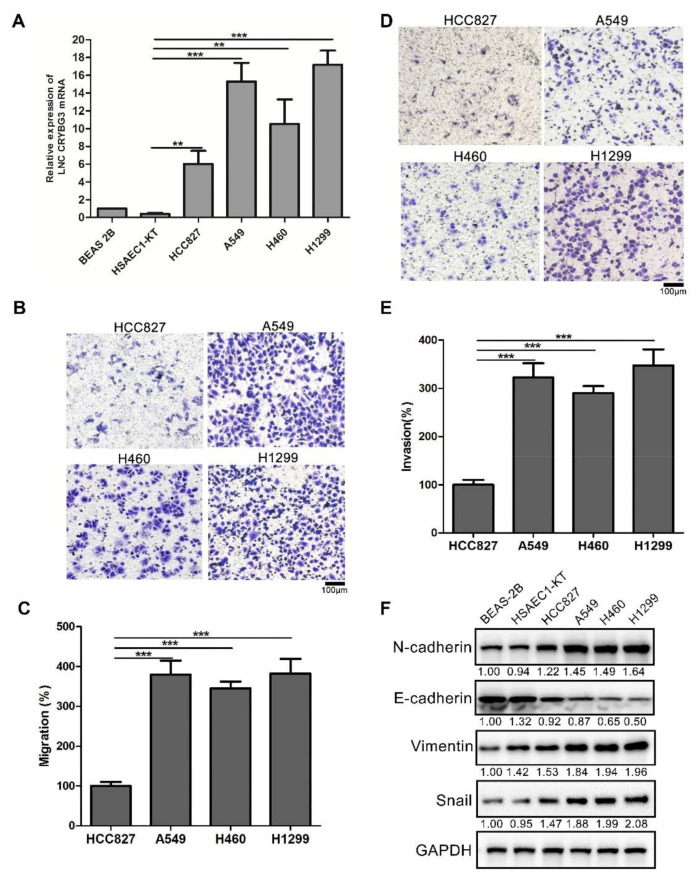
LNC CRYBG3 is specifically overexpressed in NSCLC cells. (**A**), Relative LNC CRYBG3 expression levels in the lung cancer cell lines HCC827, A549, H460, and H1299 and in the normal lung cell lines Beas-2B and HSAEC1-KT. (**B**–**E**), Migration (**B**,**C**) and invasion (**D**,**E**) activities of HCC827, A549, H460, and H1299 cells were analyzed using a Transwell assay. Scale bar: 100 μm. Data are represented as means ± SD (three independent replicates, two-tailed Student’s *t* test. ** *p* < 0.01, *** *p* < 0.001). (**F**) Western blot analysis of EMT markers (E-cadherin, N-cadherin, Vimentin, and Snail). Relative densitometry values for the representative blots are given below each band.

**Figure 2 ijms-22-03211-f002:**
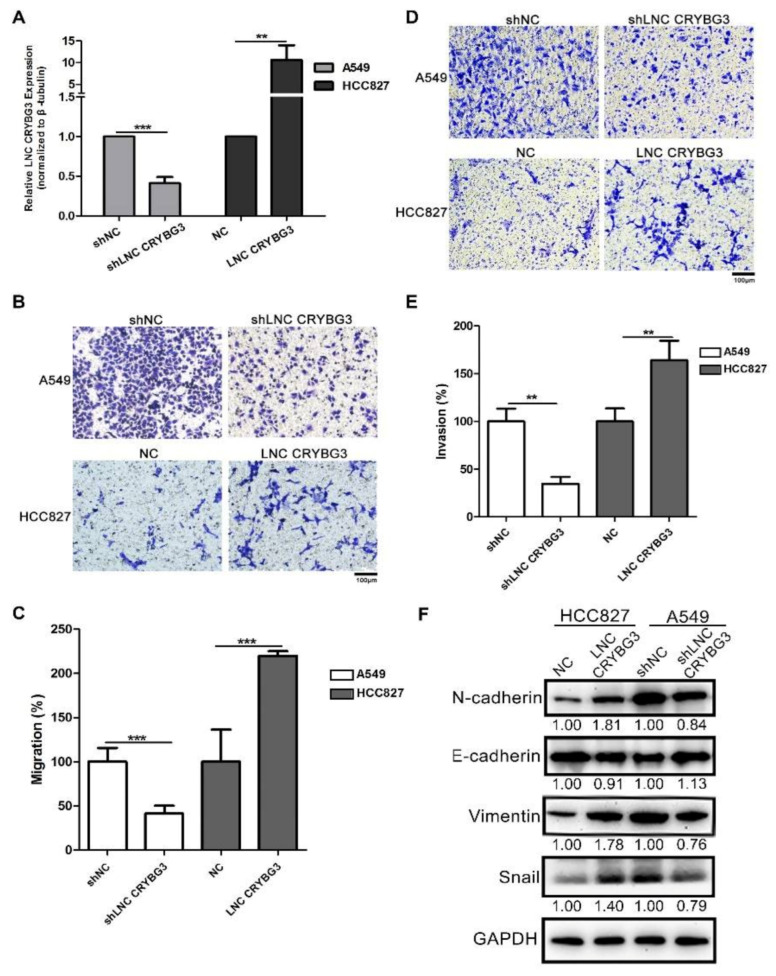
LNC CRYBG3 promotes NSCLC cell metastasis. (**A**), LNC CRYBG3 was knocked down using shRNA lentivirus particles in A549 cells (A549 shLNC CRYBG3 cells) and overexpressed in HCC827 cells using LNC CRYBG3 adenovirus particles (HCC827 LNC CRYBG3 cells). (**B**–**E**), Migration (**B**,**C**) and invasion (**D**,**E**) activities of HCC827 LNC CRYBG3, A549 shLNC CRYBG3, and control cells were analyzed using a Transwell assay. Scale bar: 100 μm. Data are represented as means ± SD (three independent replicates; two-tailed Student’s *t* test. ** *p* < 0.01, *** *p* < 0.001). (**F**) Western blot analysis of EMT markers (E-cadherin, N-cadherin, Vimentin, and Snail). Relative densitometry values for the representative blots are given below each band.

**Figure 3 ijms-22-03211-f003:**
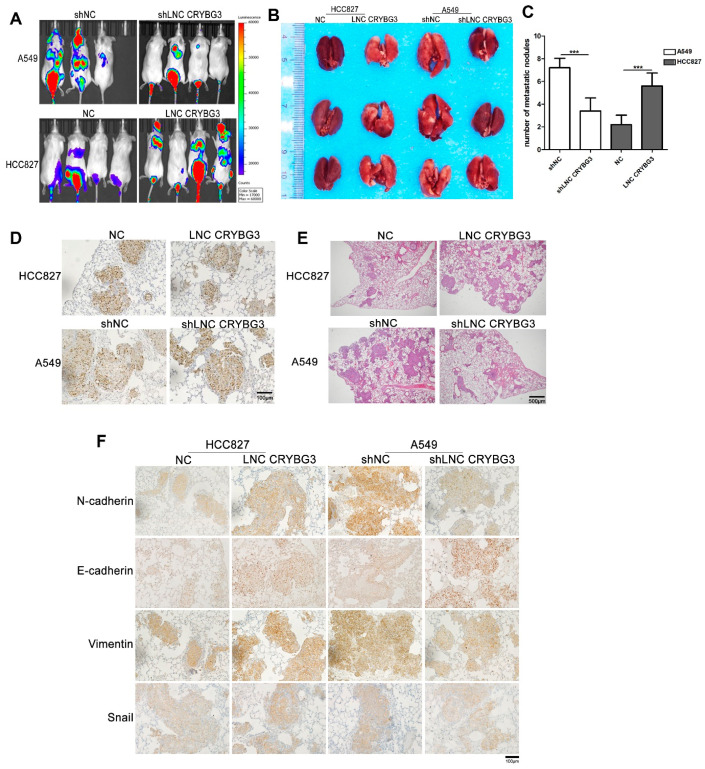
LNC CRYBG3 enhances NSCLC cell metastatic ability in NOD/SCID mice. (**A**), Images of luminescent signals in NOD/SCID mice intravenously injected with A549 shLNC CRYBG3, HCC827 LNC CRYBG3, and negative control cells with stable luciferase expression. (**B**), Representative images of lung metastatic nodules. (**C**), The number of lung metastatic nodules are represented as means ± SD using the two-tailed Student’s *t* test, *** *p* < 0.001. (**D**), Cell proliferation marker Ki67 expression detected by immunohistochemical (IHC) staining. Scale bar, 100 μm. (**E**), Hematoxylin and eosin staining sections showed lung metastases. Scale bar, 500 μm. (**F**) IHC staining showed the expression of EMT markers in the lungs. Scale bar, 100 μm.

**Figure 4 ijms-22-03211-f004:**
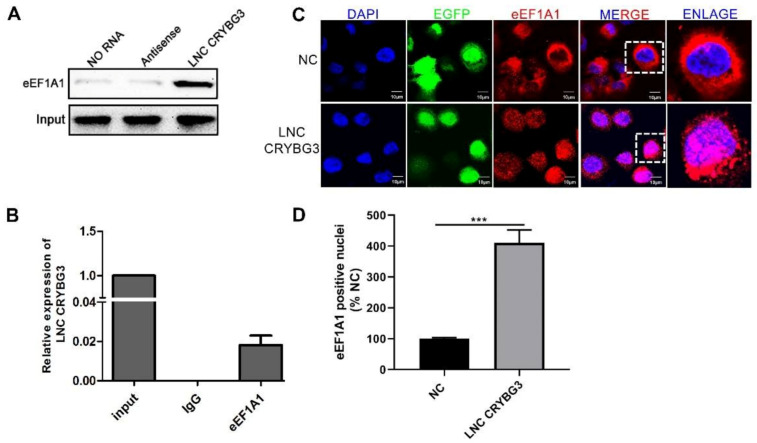
LNC CRYBG3 combines with eEF1A1 and increases its translocation to cell nuclei. (**A**), RNA pull-down analysis determined the interaction of LNC CRYBG3 and eEF1A1 in HCC827 cells. (**B**), LNC CRYBG3 enrichment histogram after RNA immunoprecipitation assays using eEF1A1 antibodies. (**C**), Immunofluorescence analysis determined the LNC CRYBG3 effect on eEF1A1 translocation. Scale bar, 10 μm. (**D**), eEF1A1 positive nuclei were quantified by counting in 6 random fields (600×). Data are represented as means ± SD (two-tailed Student’s *t* test. *** *p* < 0.001).

**Figure 5 ijms-22-03211-f005:**
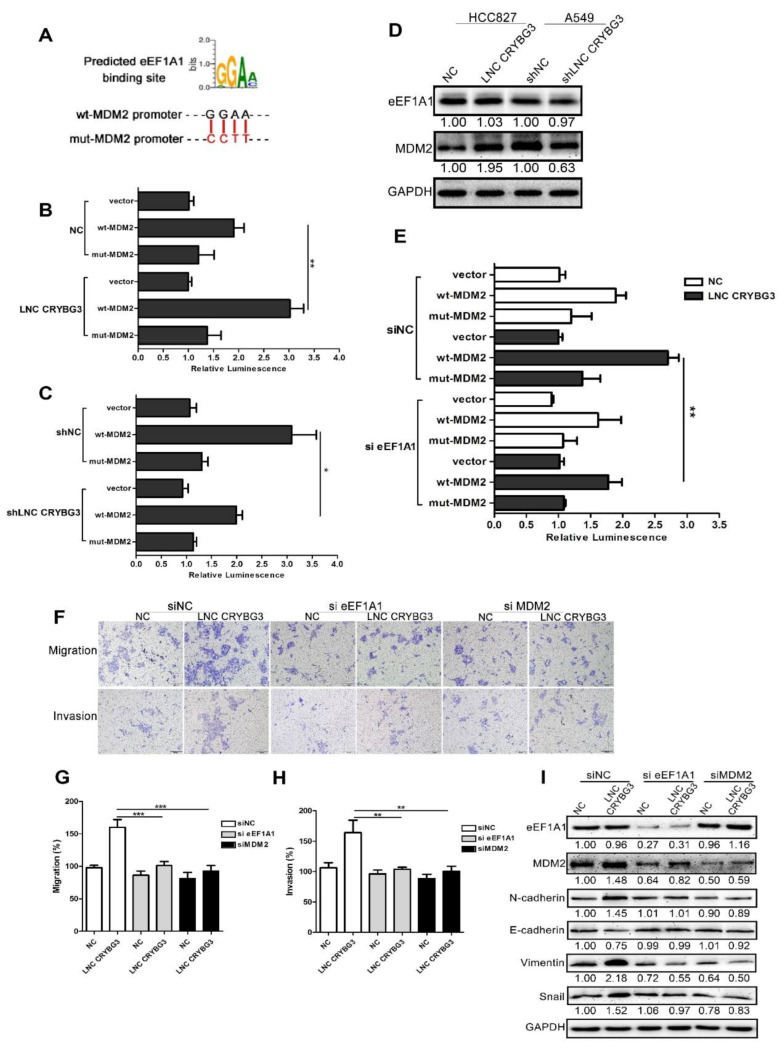
LNC CRYBG3 promotes the binding of eEF1A1 to the promoter of MDM2 gene, consequently resulting in the upregulation of MDM2. (**A**), Predicted eEF1A1 binding site in the putative MDM2 gene promoter region. (**B**,**C**), Luciferase reporter activity for the MDM2 gene promoter in HCC827 cells (**B**) and A549 cells (**C**). (**D**) Expression of eEF1A1 and MDM2 in HCC827 LNC CRYBG3 and A549 shLNC CRYBG3 cells detected by Western blotting. (**E**) Luciferase reporter activity analysis for the influence of eEF1A1 siRNA on overexpressed LNC CRYBG3-induced MDM2 expression. (**F**–**H**) Migration (**G**) or invasion (**H**) of eEF1A1 or MDM2 siRNA-treated NSCLC cells examined using a Transwell assay. Data are represented as means ± SD (three independent replicates, two-tailed Student’s *t* test. * *p* < 0.05, ** *p* < 0.01, *** *p* < 0.001). Scale bar, 100 μm. (**I**) Immunoblot analyses confirmed the effect of eEF1A1 or MDM2 siRNA on levels of EMT markers in HCC827 LNC CRYBG3 cells and negative control cells. Relative densitometry values for the representative blots are given below each band.

**Figure 6 ijms-22-03211-f006:**
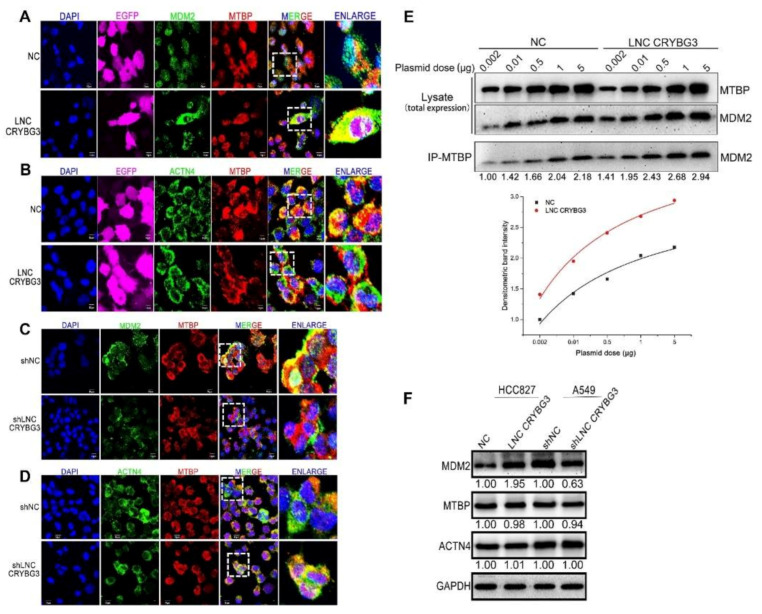
MDM2 promotes metastasis by blocking the MTBP and ACTN4 interaction. (**A**–**D**), Immunofluorescence analysis for co-localization of MDM2 and MTBP, ACTN4 and MTBP in HCC827 LNC CRYBG3, A549 shLNC CRYBG3, and negative control cells. Merged images show co-localized sites (yellow). Scale bar, 10 μm. (**E**), HCC827-NC and HCC827-LNC CRYBG3 cells were co-transfected with MDM2 and MTBP (with plasmid dose of 0.002 μg, 0.01 μg, 0.5 μg, 1 μg and 5 μg) for 24 h, then co-immunoprecipitation assay was performed using MTBP-conjugated antibody microbeads to detect the MDM2 and MTBP interaction for each transfected case. The immunoblot band intensity of the MDM2 combined with MTBP in each transfect case was normalized to NC cells with 0.002 μg plasmid (measured by image J), then relative densitometry values for the representative blots were given below each band. The data (below) was analysed by OriginPro 8.0 using hyperbolic function. (**F**) Western blotting indicating MDM2, MTBP, and ACTN4 levels in cells with LNC CRYBG3 overexpression or depletion. Relative densitometry values for the representative blots are given below each band.

**Figure 7 ijms-22-03211-f007:**
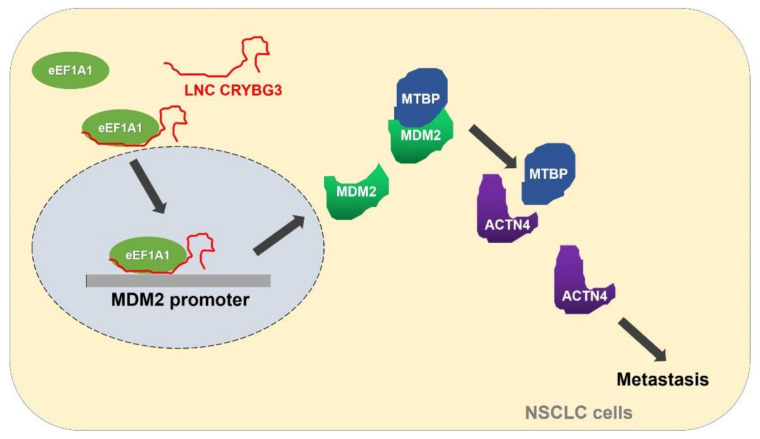
Demonstration of biological roles and pathways impacted by LNC CRYBG3 on metastasis of NSCLC cells.

## Data Availability

Data are presented in main text and Appendix A.

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
