# Peer review of "Long Non-Coding RNA CRYBG3 Promotes Lung Cancer Metastasis via Activating the eEF1A1/MDM2/MTBP Axis"

_ijms, 2021, doi:10.3390/ijms22063211_

Round 1
Reviewer 1 Report
The manuscript “Long Non-Coding RNA CRYBG3 promotes Lung Cancer Metastasis Via Activating eEF1A1/MDM2/MTBP Axis” by Wu et al. aims at revealing the mechanism by with lncCRYBG3 promotes metastasis in NSCLC. The manuscript is well written. Though they have lots of data in the support of their conclusion, I am not convinced with several points in their paper. Therefore, I do not recommend publishing this article in this journal. The major points I disagree with is:
The authors claim that MTBP and MDM2 interaction is increased in the presence of lncCRYBG3. To prove that they are using IP with controls. This is very misleading. They need to use cell-based method where the expressions of protein are clearly monitors and determine Kd value between MTBP and MDM2 in the absence and presence of lncCRYBG2. If Kd value between MTBP and MDM2 in the presence of lncCRYBG3 is large compared to the absence of lncCRYBG3 only then one can say interaction is enhanced.
Minor Points:
- Please include BEAS2B and HSAEC1-KT lines as well in your western blot in Figure 1F.
- Do this for Figure1B-1E.
- Are the p-values obtained from one way ANOVA?
Reviewer 2 Report
In this study, the author address the role of the lncRNA CRYBG3 in driving metastatic advance in non-small cell lung cancer. The topic is interesting and the findings may harbor translational potential. Techniques used are also in line with the topic.
I would suggest to revise the introduction and better explain the results in general, sometimes they are presented too quickly. I would short the introduction a little bit and focus more on the last part-results from the study. Please also be careful to statistical analysis and please provide additional quantification and improve figure legends. Also details on the experimental part should be added. Please pay attention to figure image quality.
I have some specific comments:
line 83-85 no need to go in details with the technique or number of samples
in Figure 1: could you please better explain how did you asses migration and invasion? 1B_C_D where are the control cell lines to compare with? are the values normalized? Figure 1E: quality of the western blot is really low, it seems a snapshot from another figure. Please provide a better image and a quantification. in the figure legend F is not bold and the legend is not correct or clear (western blot Data are represented as means ± SD (three independent replicates; **p < 0.01, ***p < 0.001)??
Figure 2: results from the western blot are not clear especially for E-Cadherin. Quality of the image is again really poor, especially for snail. an you provide a quantification? again the figure legend is not complete
Figure 3: could you provide data on mice survival?
Figure 4A-B: could you move to supplementary figure? 4E: could you provide better images? colors really seems fake. For the nuclear translocation, could you provide a quantification of the positive cells?
In addition, could you provide a proof by western blot, by analyzing both cytoplasmic and nuclear compartments?
page 10: why all the lines are in different font size?
Figure 6G: western blot is not representative, not in line with text.
Round 2
Reviewer 1 Report
The authors have sent responses without new data. What I asked them, they did not do that. Rather than doing actual experiments, they are trying to justify things that they have done. They needed to do the new experiment using IP/other methods:If what they are saying is correct then they need to do the following experiment. Co-transfect the cells with MDM2 and MTBP ( with total 0.002 ug, .01ug, 0.5ug, 1 ug and 5ug). You may use equal amounts of both DNA to reach the total amount. Assuming more DNA you will transfect you will get more expression, you need to probe total ( sum of intensities of MDM2 and MTBP) and then the strength of interaction for each transfected case. Thus, you will get Interaction strength vs total expression curve. Do this experiment with and without lnc CRYBG3. If you get lower value in the interaction graph without lnc RNA only then what authors said in the paper is correct otherwise it may be the artifact of concentration and the phenotype could be due to something else not due to what authors proposed. The authors may use other methods to get the interaction strength.
Round 3
Reviewer 1 Report
I am happy that authors tried this experiment. However, I have several concerns on this experiment:
- The authors ran the gel with and without LNCCRYBG3 to obtain the interaction strength as a function of transfected DNA (Concentration of protein) as shown in the graph. However, it is very misleading. I want the authors to do the following:
- Run the Gel with and without LNC CRYBG3 in the same gel not in two separate gel i.e., use 14 lane gel and in that use 5 lane for NC samples and 5 lanes with lnc CRYBG3 and then plot the graph.
- Put these figures in the main text and uncropped gel (i.e., original gels in supplementary figures).
- Interaction Strength is defined as:
∆G = - RTln(Kd)
Usually, people get binding curve for interacting partners in the absence and presence of ligand and calculate Kd.
I am not understanding why you fit the data with straight line. What this fitting is going to tell you?
I feel like what you are saying interaction strength, it is not interaction strength. It is let’s say IP value. You needed to fit this data with (Hill fit or Hyperbolic fit). That will give you Kd value in the absence of lnc.
Similarly, you fit the data in the presence of lnc with hyperbolic Fit. That will give you Kd value in the presence of lnc.
Now you need to compare the Kd value in the presence of lnc with Kd value in the absence of lnc.
If Kd value in the presence of lnc is lower than Kd value in the absence of lnc only then you can say lnc in increasing the interaction strength.
(For more information on it please look at papers from Robia lab located Stritch school of Medicine, Maywood, IL, USA and Hristova Lab located at Johns Hopkins University, Baltimore, MD, USA).
- Please include control samples in Figure 1F. Also include all the uncropped blot in supplementary figure.
The blot you provided in reviewers comment reply, you can see that whatever result you want you can get it. Pretty pictures may look pleasant to eye, but they cannot answer the deep scientific questions.
Round 4
Reviewer 1 Report
Now, I liked the manuscript. It is much better. The authors have partially addresses my previous comments. I still want to know rom them the following:
- Why did they fit the data with straigh line? What it is going to tell the reader? They disagree with me that is totally fine. But, they needed to tell me/reader what is the meaning of slope and intercepts of the fitted line? How should we interpret that.
- Could you please do one thing? You have IP vs DNA data. The hyperbolic function is given by:
V = Vmax (IP/Kd+IP).
I assume you have Origin software. Using this software fit your data to obtain Kd in the absence and presence of lnc. Please take the help of quantitative guys in your university. I am sure they will help you.
You have done great job in responding the reviewers commment. I am sure you can do that. Please take some time to figure it out and I am sure you can do that.
